# Optimizing Organic Carrot (*Daucus carota* var. sativus) Yield and Quality Using Fish Emulsions, Cyanobacterial Fertilizer, and Seaweed Extracts

Allison Wickham and Jessica G. Davis *

Department of Soil and Crop Sciences, Colorado State University, Fort Collins, CO 80523-1170, USA;
wickham.allison@gmail.com
* Correspondence: jessica.davis@colostate.edu

**Abstract:** Liquid fertilizers are often used in the middle of the growing season in an attempt to enhance organic carrot (*Daucus carota* var. sativus) yield and quality, although their effect on plant performance is unproven. The impact of liquid organic fertilizers and foliar seaweed applications on carrot yield and quality characteristics were evaluated on certified organic land at the Colorado State University Horticulture Field Research Center in Fort Collins, CO, USA, in 2014 and 2015. Hydrolyzed and non-hydrolyzed fish fertilizer and cyanobacterial fertilizer (cyano-fertilizer) treatments were applied through a drip irrigation system at prescribed N rates about every 10 days throughout the growing season. Each treatment, including the unfertilized control, was repeated with the addition of concentrated organic seaweed extract, containing phytohormones, applied foliarly at the manufacturer's recommended rates. The cyano-fertilizer treatment resulted in longer carrots in 2014 and the highest carrot yield in both years, with it consistently yielding equal to or greater than either hydrolyzed or non-hydrolyzed fish fertilizer. The foliar seaweed applications had no effect on carrot yield in either year. The cyano-fertilizer performed comparably to the other fertilizers, suggesting that cyano-fertilizer could be a viable alternative to organic liquid fish fertilizers.

**Keywords:** organic fertilizer; *Daucus carota* var. sativus; *Anabaena*; cyanobacteria; fish emulsion; seaweed





## 1. Introduction

Nitrogen (N) management on organic farms can be difficult due to the nature of certified organic fertilizers. Organic farmers use compost, manure, legume cover crops, dried organic products such as feathers and blood meal, or liquid fertilizers such as fish emulsion to increase crop productivity. Additionally, there are many specialty products on the market containing plant growth regulators or phytohormones, intended to impact plant growth characteristics to increase yield and/or quality. The use of these fertilizers is often imperfect; liquid fertilizers and meals are energy intensive to produce and ship, and the yield impacts of specialty products are often inconclusive [1–3]. Nitrogen mineralization rates of manures and meals are difficult to predict or control [4,5]. To improve the predictability of N inputs, organic farmers often turn to fertilizers such as liquid fish fertilizer to supply their crops with supplemental N mid-season. Purchasing and transporting fertilizers such as fish fertilizer can be costly and can have a large carbon (C) footprint, contrary to the aims of organic agriculture. By growing cyanobacterial fertilizer (cyano-fertilizer) organically on-farm, organic farmers can harness the N-fixing ability of these prokaryotes and potentially decrease the cost and C footprint of purchasing and transporting traditional organic fertilizers [6–8]. While cyano-fertilizer has a lower N concentration than fish fertilizers, fish fertilizers are usually diluted prior to application, and both can be applied multiple times throughout the growing season through drip irrigation systems.

Phytohormones are marketed by manufacturers to stimulate plant growth when applied externally. Organic farmers can purchase products such as liquid seaweed extract

to apply phytohormones to crop foliage or as a soil soak. Seaweed products are best known for their auxin and cytokinin contents. Studies have found that cyanobacteria can produce an elaborate array of secondary plant compounds, including auxins, cytokinins, and abscisic acid [9,10]. These compounds can affect the nutritional value and water use efficiency of plants [11,12]. Conclusions regarding the impacts of added phytohormones vary, and often the dosage and the location of application have different effects. In a study evaluating the response of tomatoes, carrots, and potatoes to a cytokinin-containing product, only the tomatoes produced a greater yield in response to the added cytokinin [1]. Seaweed concentrate was reported to increase tomato seedling growth when used as a soil soak, while tomato yield was increased through foliar application [13]. In another study, seaweed extract was applied to Russet Burbank potatoes for seven years with no increase in yield, but the Lemhi Russet variety showed a yield response in three of the five years [14]. In our study, fish emulsions and cyano-fertilizer were applied to the soil with irrigation, and seaweed concentrates were applied foliarly to better understand the impacts of each on the carrots.

Carrot (*Daucus carota* var. sativus) is a member of the Umbelliferae family [15]. Carrots have a moderate N requirement and perform well with 100 to 135 kg available N ha$^{-1}$ [15]. The timing of N application is of equal importance, as carrots must be "spoon-fed" N to prevent growth spurts and cracking [15]. Carrots can take up 72–250 kg N ha$^{-1}$ depending on the soil conditions [15]. In a dry year, net N removal for carrots was 72–81 kg ha$^{-1}$, which is lower than previously cited values of 150 kg ha$^{-1}$ in Finland and 178 kg ha$^{-1}$ in Michigan, USA [16]. In the same study, carrot yields did not respond to an N rate of 110 kg ha$^{-1}$, and carrots usually had sufficient soil N without fertilizer application [16]. Conversely, other researchers reported that the yield and quality parameters of carrots were maximized at 140–160 kg N ha$^{-1}$ depending on the planting date [17].

In this study, carrots were grown in Fort Collins, CO, USA, in the 2014 and 2015 growing seasons to evaluate the impact of liquid organic fertilizers and foliar seaweed extracts on yield and quality. The specific objectives were to (1) evaluate cyano-fertilizer compared to hydrolyzed and non-hydrolyzed liquid fish fertilizers in providing adequate N to optimize the yield and quality of carrots without leaving excessive residual inorganic N in the soil post harvest, (2) characterize the impact of foliar liquid seaweed on carrot characteristics, and (3) evaluate the potential impact of phytohormones in seaweed, fish fertilizer, and cyano-fertilizer on carrots.

## 2. Materials and Methods

### 2.1. Site Description and Planting

Field experiments were conducted during the 2014 and 2015 growing seasons on certified organic land at the Colorado State University (CSU) Horticultural Research Center (4300 E County Road 50, 80524) in Fort Collins, CO, USA. The soil in this location is a fine, smectitic, mesic Aridic Argiustoll of the Nunn series [18]. The soil pH was 8.1, and the organic matter content was 2.7% in the top 30 cm. The average maximum temperature during the growing season (1 May–30 September) was 26 °C in both 2014 and 2015, and the average minimum temperature was 18 °C in 2014 and 17 °C in 2015. There were 2.1 cm and 6.1 cm of rain during the growing season in 2014 and 2015, respectively. The plot location within the field was moved from year-to-year to avoid residual treatment effects. Pre-season inorganic soil N analyses (0–60 cm) were performed by Ward Laboratories Inc. in Kearney, NE, USA, in 2014 and by the CSU Soil, Water, Plant Testing Laboratory (Fort Collins, CO, USA) in 2015. The soils were extracted with 2 M KCl, and nitrate ($NO_3^-$-N) and ammonium ($NH_4^+$-N) were measured by automated colorimetry. The sample results were averaged to obtain the pre-season soil inorganic N value used in determining N application rates (Table 1). The target N rate to meet the carrot crop's N needs was 135 kg N ha$^{-1}$, including both the soil inorganic N and N fertilizer applied.

**Table 1.** Existing soil inorganic N conditions present before planting the carrots. The results are based on a composite sample collected from 0–60 cm in plots in Fort Collins, CO, USA, and 1.2 kg m$^{-3}$ was the assumed bulk density used to convert from mg kg$^{-1}$ to kg ha$^{-1}$. The samples were taken on 10 April 2014 and 9 May 2015.

| | Pre-Season Inorganic N | | |
|---|---|---|---|
| **Year** | **NO$_3^-$-N** | **NH$_4^+$-N** | **Total N** |
| | - - - - - - - - - - - -kg ha$^{-1}$- - - - - - - - - - - - | | |
| 2014 | 19.6 | 30.8 | 50.4 |
| 2015 | 8.5 | 50.2 | 58.7 |

Organic "Nectar" (F1) carrot (*Daucus carota* var. sativus) seeds were purchased from Johnny's Selected Seeds (Johnny's Selected Seeds, Winslow, ME, USA). This variety is said to be uniform and flavorful and 17.7–20.3 cm in length, with medium-tall tops that hold up well to leaf blight. The carrots were double planted (sub-rows) at a seeding rate of 494,000 seeds ha$^{-1}$ in 3.05 m-length rows per plot, with 7.6 cm spacing between the plants and 76 cm centers between the rows (2.3 m$^2$ plot$^{-1}$); the planting and harvest dates are shown in Table 2. Each sub-row was planted 7.6 cm away from the center of the drip tape. After emergence, the carrots were thinned to contain approximately 40 plants per 3.05 m sub-row, for a total of 80 carrots surrounding each drip tape. To minimize edge effects, the center 10 plants (5 per sub-row) were flagged for measurements as representatives of the row.

**Table 2.** Schedule of carrot (*Daucus carota* var. sativus) field activities from planting to harvest in 2014 and 2015.

| **Field Activities** | **2014** | **2015** |
|---|---|---|
| Planting | 20 May | 8 June |
| Emergence | 1 June | 28 June |
| Harvest | 6 September (109 DAP *) | 28 August (81 DAP) |

* DAP = days after planting.

*2.2. Experimental Design*

The study was designed as a randomized complete block design with a 4 × 2 factorial scheme (4 soil treatments × 2 foliar seaweed treatments) with four replications (32 plots). The following treatments were compared: four soil treatments (one control and three fertilizer treatments) with and without foliar seaweed for a total of eight treatments. The three N fertilizers used in this experiment were: cyano-fertilizer, hydrolyzed fish fertilizer, and non-hydrolyzed fish fertilizer. The cyano-fertilizer (*Anabaena* spp. cyanobacteria) was grown on-farm [19] and had an average of 23.3 mg N/kg or <1% N by weight (Total Kjeldahl N). Neptune's Harvest hydrolyzed fish fertilizer (2-4-1) and Alaska non-hydrolyzed fish fertilizer (5-1-1) were purchased from Neptune's Harvest (Gloucester, MA, USA) and Fort Collins Nursery (Fort Collins, CO, USA), respectively. In fertilizer manufacturing, the term 'hydrolyzed' generally means that the whole fish is cold processed in water and is broken down using naturally occurring enzymes, whereas non-hydrolyzed typically means heat processed and evaporated to concentrate the nutrients.

The three fertilizers varied in N concentration and were applied at equal N rates (the application dates are shown in Table 3). The N fertilizers were injected into a drip irrigation system (described below) and applied directly to the soil. The control group received no N fertilizer, and water supplied through the N treatments was calculated and supplied to the control rows the next day to equalize the water application. In 2014, Seacom PGR Organic Seaweed Concentrate (0-4-4) was purchased from Johnny's Selected Seeds (Winslow, ME, USA), but in 2015, the Colorado Department of Agriculture would not accept this product under its organic certification; therefore, in 2015, Neptune's Harvest Organic Seaweed Plant Food (0-0-1) purchased from Neptune's Harvest (Gloucester, MA, USA) was utilized

instead. Both seaweed products were chosen because the seaweed was cold processed to retain the integrity of biological molecules and because they contained no N, reducing interference with the effects of N fertilizers. The seaweed extract was applied foliarly using a backpack sprayer following the manufacturer's recommendations (Table 4).

**Table 3.** Nitrogen (N) fertilizer applications to carrots (*Daucus carota* var. sativus). The dates of the applications and the individual and season fertilization totals by treatment type are shown. Applications were made in the 2014 and 2015 growing seasons in Fort Collins, CO, USA.

| Application Dates | Cyano-Fertilizer | Hydrolyzed Fish Fertilizer | Non-Hydrolyzed Fish Fertilizer |
|---|---|---|---|
| | - - - - - - - - - - - - - - - - -kg N ha$^{-1}$- - - - - - - - - - - - - - - - - | | |
| | 2014 | | |
| 6 June 2014 | 4.3 | 4.0 | 4.5 |
| 11 July 2014 | 7.9 | 7.9 | 7.9 |
| 1 August 2014 | 9.9 | 9.9 | 9.9 |
| 11 August 2014 | 26.6 | 26.6 | 26.1 |
| 18 August 2014 | 19.7 | 19.7 | 19.3 |
| Season Total | 68.4 | 68.1 | 67.7 |
| | 2015 | | |
| 26 June 2015 | 9.7 | 9.6 | 9.7 |
| 3 July 2015 | 3.0 | 3.0 | 3.0 |
| 17 July 2015 | 7.6 | 7.6 | 7.6 |
| 23 July 2015 | 8.1 | 8.1 | 8.1 |
| 30 July 2015 | 0.0 | 10.7 | 10.7 |
| 7 August 2015 | 0.0 | 10.7 | 10.7 |
| 21 August 2015 | 0.0 | 32.2 | 32.2 |
| Season Total | 28.3 | 82.0 | 82.0 |

**Table 4.** Foliar seaweed applications to carrot (*Daucus carota* var. sativus) in the 2014 and 2015 growing seasons in Fort Collins, CO, USA. The season application dates and totals are shown. PGR seaweed is concentrated and has a greater dilution rate, hence the difference in the product volumes. Both seaweed products were applied at the manufacturer's recommended rates.

| Application Dates | Seaweed Type and Application Rates |
|---|---|
| | - - - - - - - - - - - -L ha$^{-1}$ - - - - - - - - - - - - |
| | 2014: Seacom PGR Seaweed * |
| 3 June 2014 | 1.2 |
| 8 July 2014 | 1.2 |
| 29 July 2014 | 1.2 |
| Season Total | 3.6 |
| | 2015: Neptune's Harvest Seaweed ** |
| 26 June 2015 | 31.7 |
| 3 July 2015 | 31.7 |
| 17 July 2015 | 47.5 |
| 7 August 2015 | 47.5 |
| 21 August 2015 | 63.4 |
| Season Total | 221.9 |

* Applications were made at seedling emergence and twice more at five-week and three-week intervals. ** The seaweed was diluted at 30 mL L$^{-1}$ with enough solution to coat all leaves of the plants in each foliar application. The seaweed extract was applied five times over the season, once after transplanting and thereafter at two-week intervals.

### 2.3. Irrigation System and Fertilizer Application

A drip irrigation system was installed to supply water and the N fertilizers. The system utilized two 24.4 m headers running lengthwise, outlining the plot with drip tape rows in between. A ball valve was installed at each end of the drip tape where it joined with the headers for the purpose of selectively closing the rows to facilitate fertigation

through the irrigation system. Large ball valves were installed at the ends of the headers so that low-pressure, clean water could be flushed through the lines between treatments to minimize cross-contamination. The drip tape used was 15 mil, with 20 cm spaced emitters and 4.1-L min$^{-1}$ m$^{-1}$ (John Deere, Moline, IL, USA). Irrigation was automated to run for 45 min 5 days a week. Irrigation water was not applied 2 days a week to prevent overwatering. On days that fertilizer was applied through the irrigation system, irrigation water was not applied. The applied irrigation water for the carrots totaled 82 cm and 61 cm in 2014 and 2015, respectively.

N fertilizers were applied directly through the drip irrigation system utilizing the row valves to control application to the appropriate rows. The cyano-fertilizer was grown in a production raceway on-farm and applied at full strength using a sump pump placed in the raceway [19]. Fish fertilizers were diluted into livestock watering tanks to match the cyano-fertilizer N concentration for each application and applied with a sump pump. The N content of the cyano-fertilizer was measured on the day of fertigation using a DR3900 Benchtop Spectrophotometer (Hach, Loveland, CO, USA) to measure total Kjeldahl nitrogen.

Based on the 2014 pre-season soil sampling, the fertilizer N requirement for carrots was 85 kg N ha$^{-1}$ to reach a total of 135 kg N ha$^{-1}$. Due to weather challenges, only 68 kg N ha$^{-1}$ were applied to the carrots before harvest (Table 3). In 2015, contamination of raceways with a predatory microbe negatively impacted cyano-fertilizer production. No cyano-fertilizer was applied after 23 July 2015. Therefore, cyano-fertilizer was applied at a rate of 28 kg N ha$^{-1}$, less than the recommended rate of 81 kg N ha$^{-1}$ (Table 3). All 2015 data reflect the reduced application rates for cyano-fertilizer, although the fish emulsions were applied at the full recommended rates (Table 3).

### 2.4. Harvest Sampling and Analyses

On the date of harvest, the center 10 carrots from each row were harvested, and the tops were cut off at the crown. The carrot roots were examined for deformities. The number of individual carrots displaying branched roots, root knobs, cracks or splits, and underdeveloped root lengths were counted for each row (Figure 1). The carrot root length and circumference at the crown were measured with a tape measure. Circumference was converted to diameter for statistical analysis. After harvest, each plot was soil sampled (0–45 cm) under the drip tape using a Giddings soil sampling rig (Giddings Machine Company, Windsor, CO, USA), and the samples were air-dried, ground, sieved through a 2 mm mesh, extracted using a 1:10 soil to solution ratio in 2 M KCl, and analyzed for $NO_3^-$-N and $NH_4^+$-N. In 2014, the extracts were analyzed with the auto-analyzer (Alpkem, Gorenjska, Slovenia) at the CSU EcoCore Analytical Services Laboratory in Fort Collins, CO, USA. In 2015, the extracts were analyzed by the CSU Soil, Water, and Plant Testing Laboratory in Fort Collins, CO, USA, with a Lachat (Lachat Instruments, Milwaukee, WI, USA) auto-analyzer.

Phytohormone analyses were conducted at the Proteomics and Metabolomics Facility, CSU. The fertilizer samples were adjusted to pH 7.0 with 1 N NaOH and extracted three times with water-saturated n-butanol followed by vacuum drying [20]. The extracts obtained were filtered through membrane filters (pore size 0.45 μm). Supernatants were harvested by centrifugation at 5000× *g* for 20 min at 4 °C, homogenized in liquid nitrogen using a cold mortar and pestle at 4 °C, and extracted using 80% methanol containing 10 mg L$^{-1}$ butylated hydroxytoluene at 4 °C. The samples were methylated with diazomethane and dissolved in heptane, and analysis performed with a gas chromatograph—mass spectrometer (GC-MS) [21]. The amounts of phytohormone applied over the growing season were determined by multiplying the measured concentrations by the amount of fertilizer applied (Table 5).

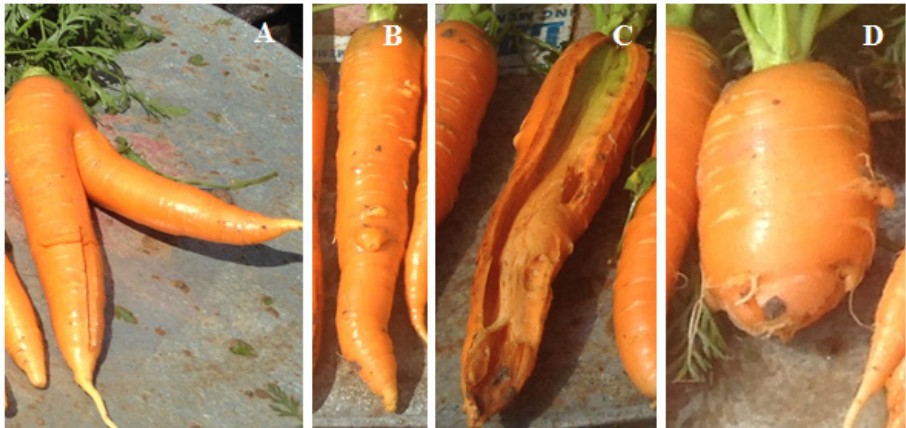

**Figure 1.** Carrot (*Daucus carota* var. sativus) root deformities: branching (**A**), root knobs (**B**), cracks or splits (**C**), and underdeveloped root length (**D**) quantified after carrot harvest in 2014 and 2015 in Fort Collins, CO, USA.

**Table 5.** Fertilizer phytohormone concentrations and application rates as measured in the fertilizers used in the 2014 and 2015 field studies in Fort Collins, CO, USA. The values were normalized for comparison based on a fertilizer application rate of 68 kg N ha$^{-1}$ (N application rate in 2014).

| Fertilizer Treatment | Phytohormone Concentrations | | | Application Rates | | |
| --- | --- | --- | --- | --- | --- | --- |
| | Auxin | Salicylic Acid | Cytokinin | Auxin | Salicylic Acid | Cytokinin |
| | - - - - - - - - - -mg kg$^{-1}$- - - - - - - - - - | | | - - - - - - - - -kg ha$^{-1}$- - - - - - - - - | | |
| Cyano-fertilizer | $6.50 \times 10^{-5}$ | $5.92 \times 10^{-3}$ | n/d | $1.88 \times 10^{-4}$ | 0.02 | n/d |
| Hydrolyzed fish fertilizer | $3.97 \times 10^{-4}$ | 0.018 | n/d | $1.22 \times 10^{-6}$ | $2.35 \times 10^{-4}$ | n/d |
| Non-hydrolyzed fish fertilizer | 1.436 | 0.077 | n/d | $1.79 \times 10^{-3}$ | $2.25 \times 10^{-5}$ | n/d |
| Seacom PGR Seaweed * | 0.802 | 48.17 | n/d | $8.75 \times 10^{-7}$ | $5.25 \times 10^{-5}$ | n/d |
| Neptune's Harvest Seaweed * | n/d ** | n/d | n/d | n/d | n/d | n/d |

* Seaweed N concentration = 0. Applied following the manufacturer's recommendations. ** n/d—none detected.

### 2.5. Statistical Analysis

All statistics were performed using Statistical Analysis Software 9.4 (SAS Institute, Inc., Cary, NC, USA). The PROC Mixed statement was used, and the experimental design was run as a $4 \times 2$ factorial scheme. The treatment and foliar seaweed applications were the fixed effects, and blocks, or replicates, were treated as a random variable. There were several factors that made it difficult to compare the 2014 and 2015 seasons; there was much more rain in 2015, and the foliar seaweed product was changed between seasons. For these reasons, the years were analyzed separately. The slice statement was used to analyze the effects of foliar seaweed extract. An adjusted F-test of fixed effects was performed using the REML method. The least-square means were estimated with the LSMEANS statement and compared with the PDIFF statement. *p*-values < 0.05 were considered significant in all cases.

## 3. Results

### 3.1. Phytohormones

Although the phytohormones in cyano-fertilizer are lower in concentration than fish fertilizers, fish fertilizers are diluted when applied, while cyano-fertilizers are applied at full strength. When all of the N fertilizers were applied at equal N rates, more salicylic acid was applied to crops from cyano-fertilizer than either of the two fish fertilizers or the two liquid seaweed products applied at the manufacturer's recommended rates (Table 5). In addition, the auxin application rate applied in the cyano-fertilizer was moderate between the non-hydrolyzed fish emulsion (the highest) and the hydrolyzed fish emulsion.

The non-hydrolyzed fish fertilizer had higher auxin and salicylic acid concentrations than the hydrolyzed fish fertilizer, but when considering the manufacturer's recommended application rates, the non-hydrolyzed fish fertilizer had a higher auxin application rate per hectare but a lower salicylic acid application rate per hectare as compared to the hydrolyzed fish fertilizer (Table 5).

The seaweed product used in 2014 (Seacom PGR) had detectable levels of auxin and salicylic acid, but no cytokinins were detected (despite the claim on the label that 400 mg kg$^{-1}$ were present). On the other hand, the seaweed product used in 2015 (Neptune's Harvest) had no detectable levels of auxin, salicylic acid, or cytokinins.

### 3.2. Yield and Quality

The nitrogen treatments significantly impacted carrot yield in both years (Table 6), but the results were not consistent between the years. Foliar seaweed had no effect on yield. In 2014, the cyano-fertilizer and the non-hydrolyzed fish fertilizer had a greater carrot yield than the control (Figure 2). In 2015, the cyano-fertilizer produced a larger carrot yield than the hydrolyzed fish fertilizer, but all treatments were comparable to the control. Although there were no significant differences in carrot diameter, the cyano-fertilizer treatment resulted in carrots with a greater average length than the unfertilized control in 2014 (Table 7). Neither the fish fertilizers nor the foliar seaweed had any effect on carrot length.

**Table 6.** F-test statistics for N treatment, foliar seaweed, and interaction effects on carrot (*Daucus carota* var. sativus) yield, post-harvest soil $NO_3$-N, root length, root knobs, and undeveloped roots.

| Variable | Yield | | Post-Harvest Soil $NO_3$-N | Root Length | Root Knobs | Undeveloped Roots |
|---|---|---|---|---|---|---|
| Year | 2014 | 2015 | 2014 | 2014 | 2014 | 2014 |
| N treatment (N) | 3.06 ** | 2.16 * | 3.78 ** | 1.71 * | 2.96 ** | 3.53 ** |
| Foliar seaweed (seaweed) | 1.97 | 0.00 | 0.00 | 0.19 | 3.84 * | 1.42 |
| N × seaweed interaction | 0.69 | 0.14 | 0.39 | 0.55 | 1.76 | 3.53 ** |

\* = $p < 0.10$, \*\* = $p < 0.05$.

**Table 7.** Carrot (*Daucus carota* var. sativus) length and average soil $NO_3{}^-$-N concentration remaining in the soil to a depth of 60 cm after harvest as influenced by fertilizer treatment in 2014. Soil $NO_3{}^-$-N was measured at the Colorado State University EcoCore Laboratory following a 2M KCl extraction.

| N Fertilizer | Average Carrot Length | Post-Harvest Soil $NO_3$-N Concentration |
|---|---|---|
| | - - - - - - - - -cm- - - - - - - - - | - - - - -mg kg$^{-1}$- - - - - |
| Control | 21.7 B * | 6.6 B |
| Cyano-Fertilizer | 23.4 A | 9.6 A |
| Hydrolyzed Fish Fertilizer | 22.8 AB | 10.0 A |
| Non-hydrolyzed Fish Fertilizer | 22.7 AB | 10.5 A |

\* Treatments that share a common letter within a column are statistically similar ($p < 0.05$).

In 2014, there were several differences in carrot root deformations as a result of the treatments (Table 6). Although all treatments produced carrots with similar incidences of cracks/splits and root branching, averaging across seaweed treatments, the hydrolyzed fish fertilizer resulted in fewer root knobs compared to the non-hydrolyzed fish fertilizer (Figure 3). There was a more pronounced difference in root knobs among the treatments when no foliar seaweed was applied. Non-hydrolyzed fish fertilizer with no seaweed applied had a higher incidence of knobs compared to cyano-fertilizer and hydrolyzed fish fertilizer treatments with no seaweed applied. Within the non-hydrolyzed fish fertilizer treatment, the addition of foliar seaweed reduced root knobs (Figure 3).

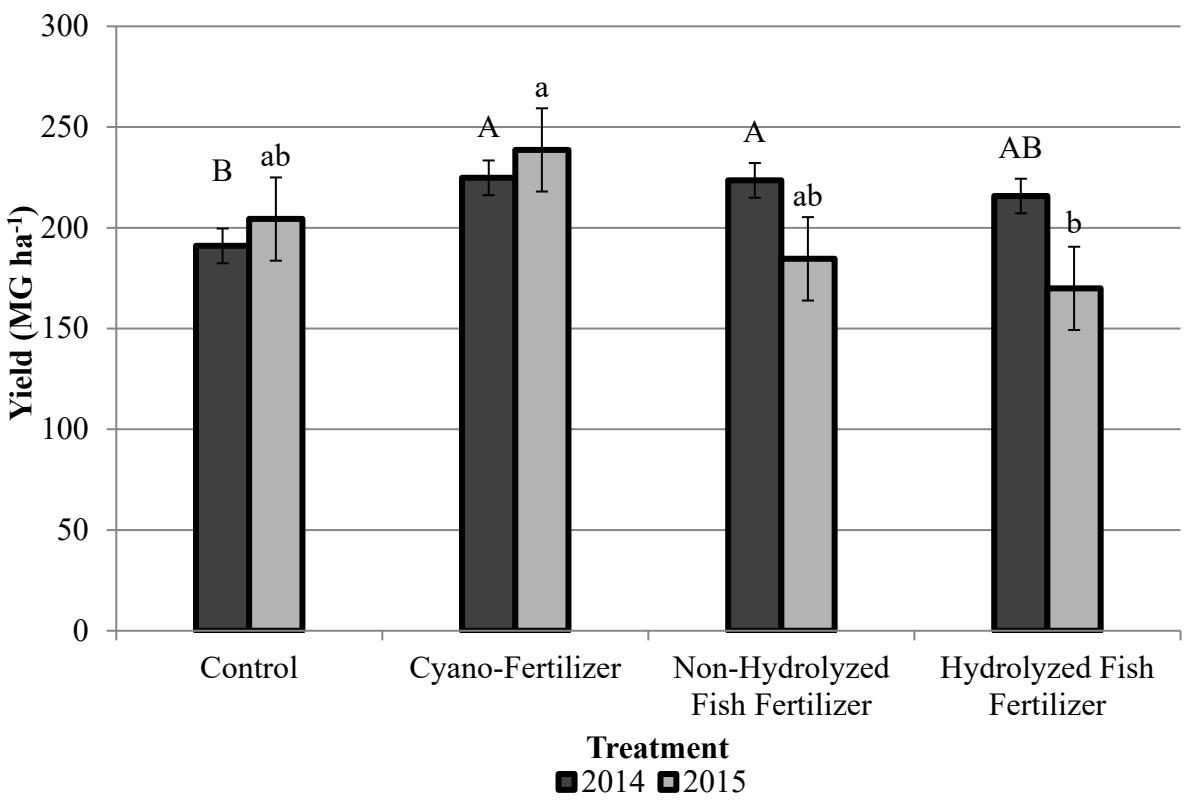

**Figure 2.** Carrot (*Daucus carota* var. sativus) yield from the field experiments in 2014 and 2015 in Fort Collins, CO, USA. Treatments that share a common letter within a year are statistically similar ($p < 0.05$ in 2014 and $p < 0.10$ in 2015).

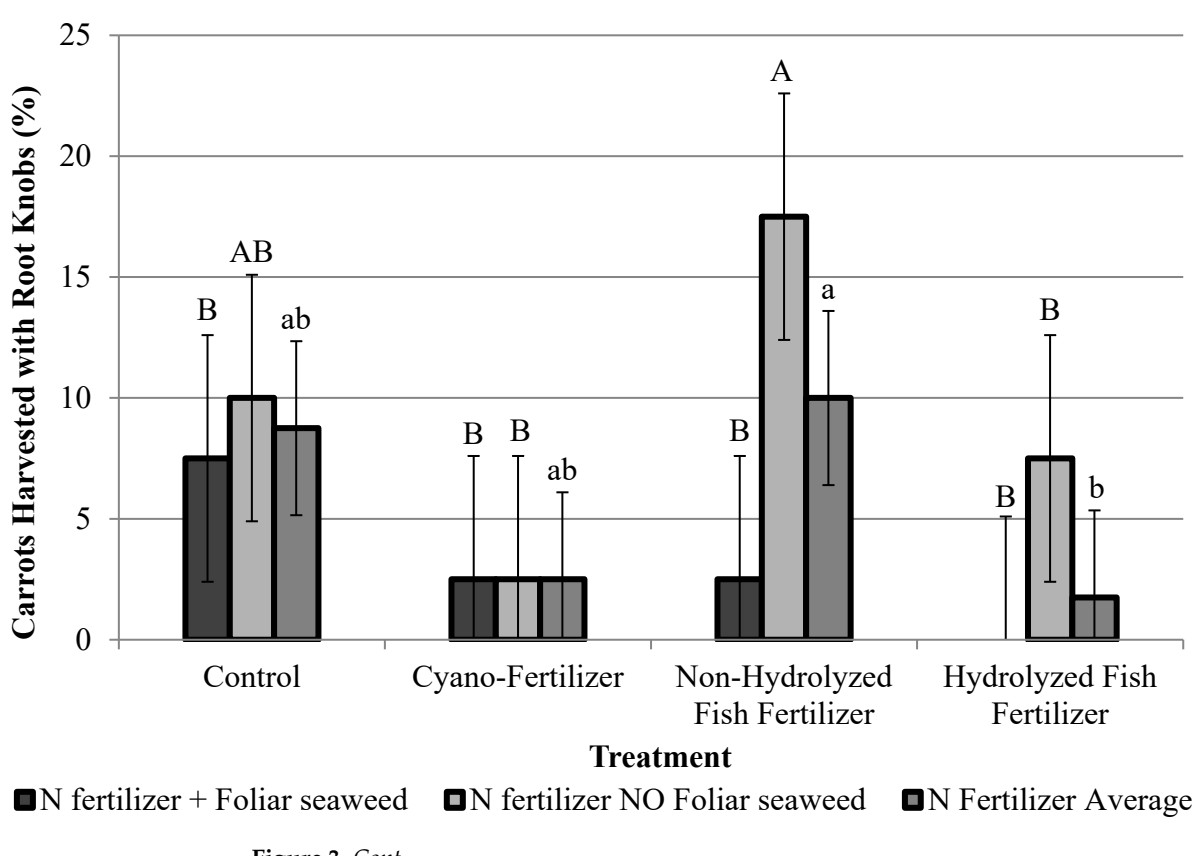

**Figure 3.** *Cont.*

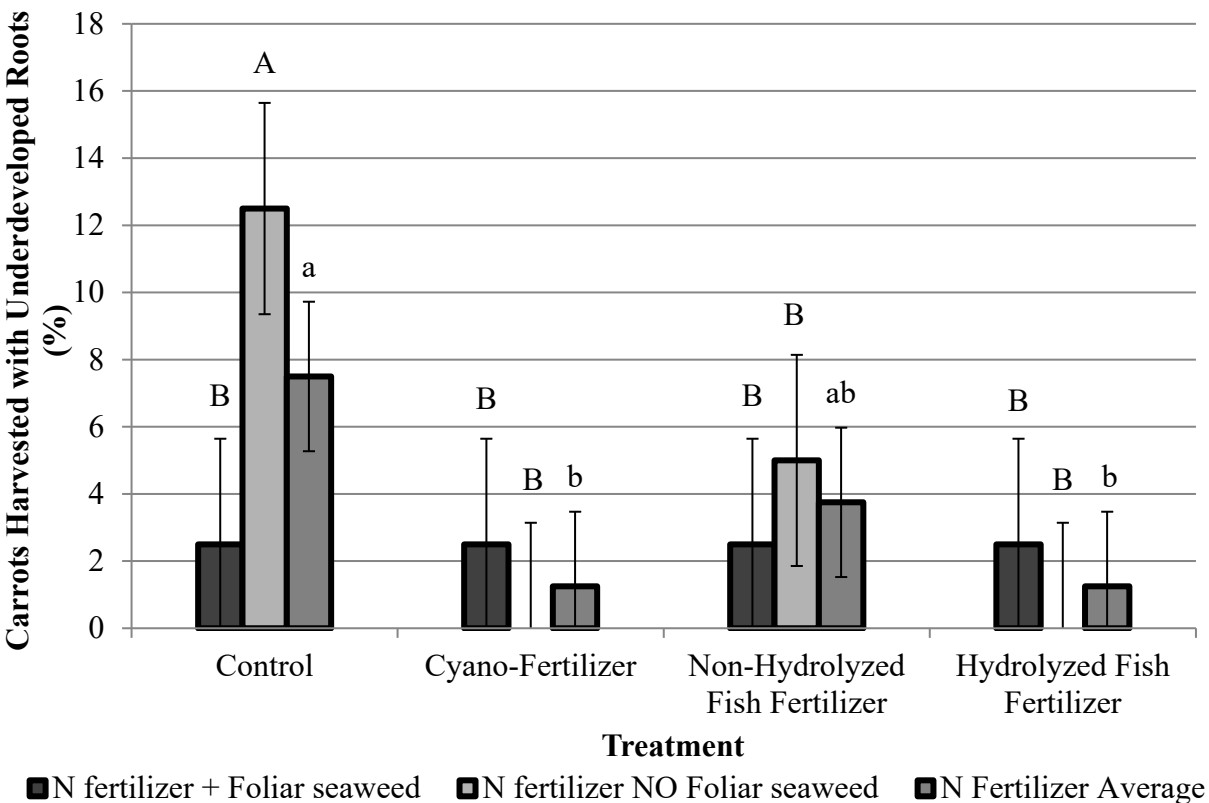

**Figure 3.** The average percentage of carrots (*Daucus carota* var. sativus) grown in Fort Collins, CO, USA, in 2014 affected by root knobs (**top**) and underdeveloped length (**bottom**) as influenced by fertilizer treatment and foliar seaweed application. Treatments that share a common capital letter are statistically similar ($p < 0.05$). Lowercase letters are used for comparison of the N fertilizer averages across the seaweed treatment.

Cyano-fertilizer and hydrolyzed fish fertilizer reduced the percent of carrots that did not fully develop in length compared to the control (Figure 3). Within the unfertilized control, the addition of foliar seaweed decreased the incidence of underdeveloped roots; however, in combination with cyano-fertilizer or fish fertilizers, foliar seaweed had no effect on the incidence of underdeveloped roots.

*3.3. Post-Harvest Soil NO$_3^-$-N*

Although there were no N fertilizer or foliar seaweed effects on plant tissue N concentration in either crop or year, there were significant N treatment effects on post-harvest soil NO$_3^-$-N concentrations in 2014, although this finding was not repeated in 2015 (Table 6). All treatments had similar post-harvest soil NH$_4^+$-N concentrations. However, in 2014, cyano-fertilizer and both fish fertilizers increased post-harvest soil NO$_3^-$-N concentrations significantly as compared to the control (Table 7).

## 4. Discussion

### 4.1. Cyano-Fertilizer

The cyano-fertilizer treatment resulted in the highest carrot yield in both years (Figure 2). The carrots fertilized with cyano-fertilizer consistently yielded equal to or greater than either fish fertilizer (even in 2015, when the N rate applied as cyano-fertilizer was much lower than the fish fertilizer treatments). In addition, the cyano-fertilizer resulted in significantly longer carrots than the control in 2014 (Table 7).

These results are consistent with other studies in which carrot quality and yield were optimized by the addition of N fertilizer throughout the season [17]. Although the results varied between the years, this study provides evidence that mid-season N applications

as cyano-fertilizer or fish fertilizer sometimes benefit carrot growth and marketability characteristics. The similarities among fertilizer treatments support the viability of cyano-fertilizer as an organic alternative to fish fertilizers.

### 4.2. Fish Fertilizers

There were no differences in carrot yield between hydrolyzed and non-hydrolyzed fish fertilizers in either year (Figure 2). However, the non-hydrolyzed fish fertilizer did produce carrots with more root knobs than the hydrolyzed fish fertilizer in 2014 (Figure 3). The hydrolyzed fish fertilizer contained higher levels of salicylic acid than the non-hydrolyzed fish fertilizer applied at equal N rates (Table 5). Salicylic acid is produced naturally in tomato plants in response to infestation of root-knot nematode, signaling plant resistance [22]. Salicylic acid can improve resistance to nematodes. If the root knots observed in this study were caused by nematodes (nematodes were not evaluated), the decrease in knot occurrence could possibly be explained by increased salicylic acid application in hydrolyzed fish fertilizer. If salicylic acid were the only compound impacting the root knobs, then one would also expect the cyano-fertilizer to have fewer root knobs than the control since it has 100-fold more salicylic acid when applied at equal N rates. This was not the case, however, as the cyano-fertilizer only showed decreased root knobs when compared to the non-hydrolyzed fish fertilizer in 2014. In the case of underdeveloped roots, the control had the most frequent occurrence of underdeveloped roots, and hydrolyzed fish fertilizer and cyano-fertilizer were significantly lower (Figure 3). These findings suggest that an unknown characteristic of the N sources may be responsible for reducing the incidence of underdeveloped roots.

Both fish fertilizers and the cyano-fertilizer increased the post-harvest soil $NO_3^--N$ concentration in 2014 (Table 7). Lower emissions of ammonia ($NH_3$) and nitrous oxide ($N_2O$) gases from cyano-fertilizer and fish emulsion as compared to blood and feather meals have been reported [23,24]. When N losses to the air are lower, one would expect more N to be stored in the soil and/or to be taken up by plants.

### 4.3. Foliar Seaweed

The foliar seaweed applications had no effect on yield in either year (Table 6). As there was only one case in which the addition of foliar seaweed reduced carrot root knobs within a treatment (non-hydrolyzed fish fertilizer), it is not clear under what conditions foliar seaweed treatments might improve carrot root quality by reducing common deformities (Figure 3). There were no significant differences in underdeveloped roots with or without foliar seaweed averaged across N fertilizer treatments. Interestingly, within the control, the addition of foliar seaweed did decrease underdeveloped roots.

This experiment is in agreement with previous findings that concentrated algae products did not increase carrot yield [1]. Perhaps seaweed products are more effective on fruiting plants (such as tomatoes) and have less impact on root crops. Because the results were inconsistent across the years, it is difficult to draw conclusions about the usefulness of foliar seaweed applications to carrots. In this study, the application of foliar seaweed products had little impact, with it having no effect on yield or quality, except for decreasing the incidence of carrot root knobs in one of the two years.

### 5. Conclusions

Cyano-fertilizer treatment resulted in the same or higher carrot yields and carrot length than the fish fertilizers in both years. The cyano-fertilizer was also equal to the fish fertilizers in % root knobs, % underdeveloped roots, and post-harvest soil $NO_3^--N$ concentration. Therefore, cyano-fertilizer is a viable alternative to liquid fish fertilizers.

Foliar seaweed applications had no significant impact on carrot yield in either year. However, in one of the two study years, foliar seaweed application decreased root knobs when applied with non-hydrolyzed fish fertilizer and decreased underdeveloped roots in the no fertilizer control.

Although the cyano-fertilizer, both fish fertilizers, and one of the seaweed products contained auxin and salicylic acid, none contained measurable cytokinin concentrations. Salicylic acid can improve resistance to root-knot nematodes, and this may possibly have been related to the knob formation on the roots. However, this requires further evaluation prior to drawing a conclusion regarding salicylic acid's impact on carrot root knobs.

**Author Contributions:** Conceptualization, A.W. and J.G.D.; methodology, A.W. and J.G.D.; investigation, A.W.; resources, J.G.D.; writing—original draft, A.W.; writing—review and editing, J.G.D.; supervision, J.G.D. All authors have read and agreed to the published version of the manuscript.

**Funding:** This research was funded by the USDA Western Sustainable Agriculture Research and Education Program project #SW14-023.

**Institutional Review Board Statement:** Not applicable.

**Informed Consent Statement:** Not applicable.

**Data Availability Statement:** Data are available upon request.

**Conflicts of Interest:** The authors declare no conflict of interest.

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
