# Peer review of "Optimizing Organic Carrot (Daucus carota var. sativus) Yield and Quality Using Fish Emulsions, Cyanobacterial Fertilizer, and Seaweed Extracts"

_agronomy, doi:10.3390/agronomy13051329_

Round 1

Reviewer 1 Report

Dear Authors,

the manuscript presents interesting results of a practical importance.

The design of the experiment is sufficient to draw conclusions, but it is desirable that its duration be longer.

There is one comment: the taxonomic name of the soil should be Italic.

The manuscript no. agronomy-2374602, titled “Optimizing Organic Carrot Yield and Quality Using Fish 2 Emulsions, Cyanobacterial Fertilizer, and Seaweed Extracts” reports the results of application of three types of liquid fertilizers, namely, hydrolysed and non-hydrolysed fish fertilizer as well as cyanobacterial fertilizer, on yield and quality of carrot. The article objective is of practical importance for organic carrot production. The authors revealed that in contrast to some earlier findings, foliar seaweed application had no effect on carrot yield, while cyanobacterial fertilizer could be a viable alternative to organic liquid fish fertilizers.

The manuscript is relevant to the topic. Some gaps in knowledge were identified. Citing literature is appropriate. Manuscript is clearly written and well structured. About 30% of the cited references are within the last 5 years. About 40% of the cited references are within the last 10 years.

There are few comments:

1. Lines 116-118: “The carrots were double planted (sub - rows) at a seeding rate of 494,000 seeds ha-1 in a 3.05 m length of row per plot, with 7.6 -cm spacing between plants and 76 -cm centers between rows (Table 2)”.

Table 2 presents only the dates of field operations, but not the information given in the sentence 119-118. (or may be in the PDF version I received the Table 2 was not fully shown)

2. The legend for Table 5 is placed under Table 6.

Author Response

Thank you for your review. We have replied to your comments in italics below.

The manuscript presents interesting results of a practical importance.

The design of the experiment is sufficient to draw conclusions, but it is desirable that its duration be longer.—Yes, we agree that a longer study would increase confidence in the results.

There is one comment: the taxonomic name of the soil should be Italic.—Ok, we have converted the soil’s taxonomic name to italics.

The manuscript no. agronomy-2374602, titled “Optimizing Organic Carrot Yield and Quality Using Fish 2 Emulsions, Cyanobacterial Fertilizer, and Seaweed Extracts” reports the results of application of three types of liquid fertilizers, namely, hydrolysed and non-hydrolysed fish fertilizer as well as cyanobacterial fertilizer, on yield and quality of carrot. The article objective is of practical importance for organic carrot production. The authors revealed that in contrast to some earlier findings, foliar seaweed application had no effect on carrot yield, while cyanobacterial fertilizer could be a viable alternative to organic liquid fish fertilizers.

The manuscript is relevant to the topic. Some gaps in knowledge were identified. Citing literature is appropriate. Manuscript is clearly written and well structured. About 30% of the cited references are within the last 5 years. About 40% of the cited references are within the last 10 years.

There are few comments:

  1. Lines 116-118: “The carrots were double planted (sub - rows) at a seeding rate of 494,000 seeds ha-1 in a 3.05 m length of row per plot, with 7.6 -cm spacing between plants and 76 -cm centers between rows (Table 2)”.

Table 2 presents only the dates of field operations, but not the information given in the sentence 119-118. (or may be in the PDF version I received the Table 2 was not fully shown)—We added some text to clarify what is shown in Table 2.

  1. The legend for Table 5 is placed under Table 6.—Thank you for pointing that out. We have corrected it.

Reviewer 2 Report

The manuscript entitled “Optimizing Organic Carrot Yield and Quality Using Fish Emulsions, Cyanobacterial Fertilizer, and Seaweed Extracts” reports the results of a two-year investigation, having the objective to evaluate the effects of organic liquid fertilizers on carrot performance and soil properties. To this aim a field experiment was carried out to study the impact of cyano-fertilizer, hydrolyzed and non-hydrolyzed liquid fish fertilizers on plants and soil, and the effects of foliar liquid seaweed on carrots characteristics.

The manuscript in several parts is difficult to understand. Then it needs to be carefully reviewed.

The fertilization is an essential agronomic practice that has the objective to improve the crop performance. But the massive use of fertilizers and the poor management could cause serious problems not only to the crop but also to the environment. Then, identifying the most appropriate fertilization technique in order to improve production and protect the environment is of great interest. The authors provided an in-depth overview of the topic and clearly explained the objective of the study. The materials and methods paragraph is long and dispersive, and too many useless details are included. In my opinion, this paragraph should be shortened and divided into sub-paragraphs for easier reading and understanding. Moreover, the presentation of results is not very clear, tables and figures should be checked. Finally, the authors provided limited discussions for some important contents/results.

The article has some weak point in methodology and presentation and discussion of results. Below I have provided some comments and remarks on the text. 

At page 1, in the abstract, it is not very clear what the authors mean with “mid-season”, explain better;

At page 2, line 49, add “growing” before “season”;

Lines 55-58, the sentence is too long; I suggest breaking this long sentence and adding a full stop after “acid [9,10]”; then the next sentence can begin as “These compounds can..”;

Lines 67-69, the sentence should be moved at the end of the paragraph, after the description of the objective. Moreover, I think that the sentence should be modified: “In our study, fish emulsions and cyano-fertilizer were applied to the soil with irrigation, and seaweed concentrates were applied foliarly …”. “Soil soak” means that the soil was become thoroughly wet by immersing it in liquid, I think that it is not the case;

Page 3, line 109: It is not very clear what the authors mean by 135 kg N ha-1. Is it the N need of the carrot during the whole cropping cycle? Explain better;

Line 126-131, the authors should better explain the experimental design and the treatments: “Two separate randomized complete block designs with a 4x2 factorial scheme (soil treatments x foliar seaweed) with four replications were adopted. The following treatments were compared: four soil treatments (control and 3 fertilizer treatments) with and without foliar seaweed for a total of 8 treatments. The three N fertilizers used in this experiment were: cyano -fertilizer, hydrolyzed fish fertilizer, and non -hydrolyzed fish……”;

At page 4, lines 183-184, modify the sentence: “Based on 2014 pre -season soil sampling, the N amount to apply during carrot growing season was 85 kg N ha-1 to reach a total of 135 kg N…”;

At page 5, lines 190-192, the sentence is not very clear, explain better;

Line 227, add “scheme” after “factorial”.

The authors should better present the results: in several tables and figures the captions and footnotes are not very clear. In the figure 1, the caption should be changed: “Carrot root deformities: branching (A), root knobs (B), cracks or splits (C), and underdeveloped root length (D).”

The footnotes of the table 5 were inserted below table 6 and the symbols used are the same as those used to indicate significance, so the authors should choose different simbols. Moreover, at line 412, the authors indicated as significance level p<0.10. Is it correct? Since, conventionally, 5%, 1% and 0.1% levels are used.

In the caption of figure 2, the sentence “Average yield of carrots in MT ha-1 was calculated by quantifying kg producedin row and between rows surrounding the center 10 carrots.” could be deleted, becouse it is explained in the text;

In the Figure 2 and in the table 7, the authors did not indicate the significance level (p = 0.005, 0.001, 0.0001);

In figure 3 (top and bottom), N Fertilizer average appears, but the authors did not explain in the text how they calculated these values and what mean.

In the above list, I attempted to indicate the main and more noticeable concerns, in any case an overall and carefully check is necessary also to improve readability and understanding.  

Decision: I believe that the paper should be published after major revisions, predominantly in methodology and presentation and discussion of results. 

Author Response

Thank you for your thorough review. We have responded to your comments in italics below.

The manuscript entitled “Optimizing Organic Carrot Yield and Quality Using Fish Emulsions, Cyanobacterial Fertilizer, and Seaweed Extracts” reports the results of a two-year investigation, having the objective to evaluate the effects of organic liquid fertilizers on carrot performance and soil properties. To this aim a field experiment was carried out to study the impact of cyano-fertilizer, hydrolyzed and non-hydrolyzed liquid fish fertilizers on plants and soil, and the effects of foliar liquid seaweed on carrots characteristics.

The manuscript in several parts is difficult to understand. Then it needs to be carefully reviewed.

The fertilization is an essential agronomic practice that has the objective to improve the crop performance. But the massive use of fertilizers and the poor management could cause serious problems not only to the crop but also to the environment. Then, identifying the most appropriate fertilization technique in order to improve production and protect the environment is of great interest. The authors provided an in-depth overview of the topic and clearly explained the objective of the study. The materials and methods paragraph is long and dispersive, and too many useless details are included. In my opinion, this paragraph should be shortened and divided into sub-paragraphs for easier reading and understanding. Moreover, the presentation of results is not very clear, tables and figures should be checked. Finally, the authors provided limited discussions for some important contents/results.—We divided the Materials and Methods into sub-sections as requested. We have also revised other sections as suggested by this reviewer and two additional reviewers.

 The article has some weak point in methodology and presentation and discussion of results. Below I have provided some comments and remarks on the text. 

At page 1, in the abstract, it is not very clear what the authors mean with “mid-season”, explain better;--We clarified this by changing the text to “middle of the growing season” (see line 9).

 At page 2, line 49, add “growing” before “season”;--We have corrected this as suggested.

Lines 55-58, the sentence is too long; I suggest breaking this long sentence and adding a full stop after “acid [9,10]”; then the next sentence can begin as “These compounds can..”;--We made this change as suggested.

 Lines 67-69, the sentence should be moved at the end of the paragraph, after the description of the objective. Moreover, I think that the sentence should be modified: “In our study, fish emulsions and cyano-fertilizer were applied to the soil with irrigation, and seaweed concentrates were applied foliarly …”. “Soil soak” means that the soil was become thoroughly wet by immersing it in liquid, I think that it is not the case;--We have changed this sentence as recommended.

Page 3, line 109: It is not very clear what the authors mean by 135 kg N ha-1. Is it the N need of the carrot during the whole cropping cycle? Explain better;--We have clarified this as requested (lines 110-112).

Line 126-131, the authors should better explain the experimental design and the treatments: “Two separate randomized complete block designs with a 4x2 factorial scheme (soil treatments x foliar seaweed) with four replications were adopted. The following treatments were compared: four soil treatments (control and 3 fertilizer treatments) with and without foliar seaweed for a total of 8 treatments. The three N fertilizers used in this experiment were: cyano -fertilizer, hydrolyzed fish fertilizer, and non -hydrolyzed fish……”; This section has been revised (lines 130-136).

At page 4, lines 183-184, modify the sentence: “Based on 2014 pre -season soil sampling, the N amount to apply during carrot growing season was 85 kg N ha-1 to reach a total of 135 kg N…”; We have modified this as requested (lines 191-193).

At page 5, lines 190-192, the sentence is not very clear, explain better; We have clarified the sentence as requested (lines 231-232).

Line 227, add “scheme” after “factorial”.—We added the word “scheme” as requested (line 238).

The authors should better present the results: in several tables and figures the captions and footnotes are not very clear. In the figure 1, the caption should be changed: “Carrot root deformities: branching (A), root knobs (B), cracks or splits (C), and underdeveloped root length (D).”—We changed the Figure 1 caption as suggested.

 The footnotes of the table 5 were inserted below table 6 and the symbols used are the same as those used to indicate significance, so the authors should choose different simbols.—Thank you for pointing this out. We have moved the Table 5 footnotes to immediately follow Table 5.

 Moreover, at line 412, the authors indicated as significance level p<0.10. Is it correct? Since, conventionally, 5%, 1% and 0.1% levels are used.—One asterisk represents p<0.10, and two asterisks represent p<0.05. In field studies, sometimes 0.10 is used, due to increased variability.

In the caption of figure 2, the sentence “Average yield of carrots in MT ha-1 was calculated by quantifying kg produced in row and between rows surrounding the center 10 carrots.” could be deleted, becouse it is explained in the text;--We removed this sentence as suggested.

In the Figure 2 and in the table 7, the authors did not indicate the significance level (p = 0.005, 0.001, 0.0001);--We overlooked this in Figure 2 and have now added the p-values into that figure. In Table 7, the p-value was given in the Table heading; we removed it from the heading and added it to the footnote.

 In figure 3 (top and bottom), N Fertilizer average appears, but the authors did not explain in the text how they calculated these values and what mean.—The Figure 3 caption states that the averages were calculated across seaweed treatment.

In the above list, I attempted to indicate the main and more noticeable concerns, in any case an overall and carefully check is necessary also to improve readability and understanding.—The authors have re-read the entire manuscript carefully and followed the guidance of all three reviewers as requested.  

Reviewer 3 Report

This study was designed to test the impacts of different organic fertilizers and seaweed extracts on carrot yield. The authors conducted a two-year field experiment and found that different nitrogen fertilizers significantly influenced yield, but seaweed showed no impacts. The results could be useful for organic farmers to select fertilizers.

The study was nicely designed and implemented. Data analysis was adequate. Results were clearly presented. I don’t have major concerns. But some details in the experimental design and implementation were missing. The experimental design was clearly but there is a lack of plot size and number of plots used in the study. F test results (ANOVA) were not presented for some quality variables. Tables 3 and 4 were not experimental results and could be moved to supplemental documents. I recommend Minor revision.

Specific comments:

L126: two separate randomized complete block designs: 1) What did the authors mean two designs? 2) It is not clear how many plots were set for the experiments. The authors need to provide more information on the design and implement of the experiments such as the number of plots, size of plot.

L226: The PROC Mixed procedure was used

Table 7: How about ANOVA (F test) results of carrot length as well as two variables in Fig. 3?

Fig. 3: It seems that three bars are missing.

Author Response

Thank you for your helpful comments. We have responded below in italics.

This study was designed to test the impacts of different organic fertilizers and seaweed extracts on carrot yield. The authors conducted a two-year field experiment and found that different nitrogen fertilizers significantly influenced yield, but seaweed showed no impacts. The results could be useful for organic farmers to select fertilizers.

The study was nicely designed and implemented. Data analysis was adequate. Results were clearly presented. I don’t have major concerns. But some details in the experimental design and implementation were missing. The experimental design was clearly but there is a lack of plot size and number of plots used in the study. –Plot size was inserted into line 118, and plot number has been inserted in line 130.

F test results (ANOVA) were not presented for some quality variables. --The F test results for root length, root knobs, and undeveloped roots have been inserted into Table 6.

Tables 3 and 4 were not experimental results and could be moved to supplemental documents.—We feel that these tables are essential to understanding the experimental design in the Materials and Methods and prefer to leave them in the text.

I recommend Minor revision.

Specific comments:

L126: two separate randomized complete block designs:

1) What did the authors mean two designs? Thank you for pointing this out. We changed the text to “a randomized complete block design” in lines 127-128. The two years were analyzed separately, and we clarified this in lines 231-234.

2) It is not clear how many plots were set for the experiments. The authors need to provide more information on the design and implement of the experiments such as the number of plots, size of plot.-- Plot size was inserted into line 118, and plot number has been inserted in line 130.

L226: The PROC Mixed procedure was used—We corrected this in line 228.

Table 7: How about ANOVA (F test) results of carrot length as well as two variables in Fig. 3?—The F test results for root length, root knobs, and undeveloped roots have been inserted into Table 6.

Fig. 3: It seems that three bars are missing.—These bars represent very low values, which are essentially zero. That is why they appear to be missing.